# Investigating the moments of "aha" and "hmm" through acoustic analysis of voice and speech in pre-service physics teacher education–A novel method for identifying significant learning moments

Kati Järvinen[1]*, Anne-Maria Laukkanen[2]☯, Anna-Leena Kähkönen[1]☯, Pasi Nieminen[1]☯, Terhi Mäntylä[1]☯

1 Department of Teacher Education, University of Jyväskylä, Jyväskylä, Finland, 2 Speech and Voice Research Laboratory, Faculty of Social Sciences, University of Tampere, Tampere, Finland

☯ These authors contributed equally to this work.

* kati.h.jarvinen@jyu.fi

**Data Availability Statement:** All relevant numerical data are within the manuscript and its Supporting

## Abstract

The aim of the study was to find whether certain meaningful moments in the learning process are noticeable through features of voice and how acoustic voice analyses can be utilized in learning research. The material consisted of recordings of nine university students as they were completing tasks concerning direct electric circuits as part of their course of teacher education in physics. Prosodic features of voice—fundamental frequency (F0), sound pressure level (SPL), acoustic voice quality measured by LTAS, and pausing—were investigated. The results showed that instances of confusion and understanding were manifested in acoustic parameters. F0 was significant in characterizing the both kind of learning instances. Confusion had lower SPL and alpha ratio, indicating that voice quality was softer than in understanding. Degree of voice pauses was lower in understanding, suggesting less hesitation or need for clarification for understanding compared to confusion. Voice research adds to the research of learning as speaker´s voice is affected by the different instances in the process of learning. This research approach can be used for identification of important instances of learning and directing these instances to closer analysis of content or interaction to further understand the learning processes. Therefore, this study is a novel contribution to the study of learning as it adds acoustic voice and speech analyses to the discipline.

## Introduction

### The "aha" and "hmm" moments in learning situations

Problem-based learning is a type of pedagogical setup used by teachers, and a popular approach in science education [e.g., 1]. In problem-based learning, the learners orient and analyse the problem, engage in self-directed problem solving, and end with a reporting activity

Information files. The voice samples are not available since they contain personal data.

**Funding:** This study was supported by the Academy of Finland through Grant 341558 (to TM). https://www.aka.fi The funder had no role in study design, data collection and analysis, decision to publish, or preparation of the manuscript.

**Competing interests:** The authors have declared that no competing interests exist.

such as explaining their answer to teachers or peers. Among its other goals, problem-based learning is used to produce developments in group work skills and self-directing skills, which are viewed as the recognition of missing knowledge and the ability to reapply the existing knowledge in a new, flexible way. Problem based learning is a very suitable context to study explanations and restructuring of knowledge [1, 2].

In the process of problem solving, people restructure available information. This can lead to sudden emergence of new understanding, insights. This kind of substantive reorganization of knowledge structure has been dubbed radical conceptual change (as opposed to incremental developments) [3, 4]. An insight has explanatory power, opposed to a single fact, as it addresses the "why" or "how" of a situation, not only the "what", and it can indicate a conflict between the current understanding and intuition. Insights are also explained as being information that can restructure previous assumptions and can result in an "aha" experience [5]. The "aha" experience involves positive emotions, the release of tension upon resolving the situation, and recognizes the impasse that was before [6].

Gopnik [7] argues that explanation is a goal-directed human activity which combines some of the properties of cognitive and motivational phenomenology. Explanation acts as the phenomenological mark of a cognitive process. The phenomenology contains recognizing that an explanation has been reached and the search for explanation. Gopnik calls these instances the "aha"-moments and the "hmm"-moments. The "aha" can be described as the feeling when a causal representation is set either by applying the theory or by revising it, while the "hmm" is indicating the feeling when evidence to which the theory has not yet set a causal representation is presented. Gopnik discusses the "aha" moments in context of explanation. In more general the "aha" moment or insight refers to a sudden moment of comprehension, realization or understanding e.g., in the context of problem solving [8, 9]. Often before the occurrence of the "aha" moment, the elements of the problem or the situation have been restructured [8], and the "aha" moment often involves positive emotions [5, 6]. There is also evidence that explanation, knowledge, or solutions obtained through "aha" moments are better remembered [6]. Also, the "hmm"-moments compel learners to resolution and action and can, then, act as a driving force to develop learner´s conceptual understanding [7, 10, 11].

In general, Pekrun et al. [12] have classified academic emotions into four categories: 1. achievement emotions which are emotions that have their stimuli or object focusing on success or achievement, 2. topic emotions relating to the actual topic being studied, 3. social emotions which relate to social relationships between students and teachers or among peers in educational context, and 4. epistemic emotions which relate to the learning process. Tyng et al. [13] also pinpoint the importance of emotions in the learning process. The epistemic emotions focus on knowledge or knowledge construction and they can be e.g. surprise, curiosity, confusion, or boredom and these epistemic feelings occur in situations of conflicting information where a new understanding emerges [14]. According to Pearce et al. [5] the definition of insight, the "aha" moment, is summarized in four key characteristics: subjectivity, suddenness, certainty, and emotions. They argue that the insights have been emphasized by their cognitive nature when they should include affective qualities of insights, too. Thus, the affective moments of insights are certain kind of epistemic emotions.

Vilhunen et al. [14] underline the importance of epistemic emotions in educational settings and the complexity of the interplay between cognitive and affective factors in learning situations. And Schneider et al. [15] suggest that generating understandings about emotional experiences in scientific sensemaking, students´ learning, interest and engagement can be enhanced. Tyng et al. [13] summarize the studies of changes in emotional states to three: 1. subjective approaches investigating subjective feelings and experiences, 2. behavioral

investigations focusing on gestural changes and facial and vocal expressions, and 3. objective approaches via physiological responses.

Since students' emotions and learning processes are intertwined [16], it is justifiable to identify these "aha" and "hmm" moments in learning. The "aha" moments are insights that are sudden moments of understanding something or realization how to move forward from cognitive impasses [8]. "Hmm" moments are moments of cognitive impasses or confusion [10]. These instances are referred to as moments of confusion and understanding in this study. The moments of understanding or confusion have previously been detected or identified for example from affective responses, such as facial expressions [7] or through neuroimaging methods [8], or students' self-evaluations [5, 6, 10, 16].

In the study of learning and interaction in learning situations, video analysis is today a vastly used method to find significant moments in the learning process [see e.g., 17, 18] and for revealing the behavioral changes. A common approach is to transcribe the talk and nonverbal information and work with this intermediate data. From this approach, it is not easy to decipher the emotions embedded in the process. Also, as Lodge et al. [11] point out, when face-to-face interaction is not present teachers can have difficulties in detecting students´ emotional responses, especially confusion. They argue that means for detecting emotions that students experience while learning are necessary. As online and remote learning grows the methods for detecting students´ emotions is becoming more essential. Tyng et al. [13] show the evidence of emotion modulating attention and retrieval of information through a review of neuroimaging studies and conclude their review with a call for education research to pick up the ball. However, neuroimaging techniques are impractical for classroom research. The goal of deciphering the interplay of emotion and learning progresses requires new methodologies. This present study focuses on the emotions in the learning process, the "aha" and "hmm" moments, as it aims to clarify whether these moments can be studied and recognized with acoustic analyses.

## Verbal and non-verbal communication

Interpersonal communication has traditionally been divided into two: verbal and non-verbal communication [19], and classroom talk is traditionally investigated through analyzing teacher-student speech and interaction by linguistic ethnography and sociocultural research [20]. However, linguistic content and representations do not take the individual characteristics of voice into account and sufficiently represent the paralinguistic information in speech. A textual representation of a spoken utterance can, therefore, be an insufficient indicator of both its content and its functional intent. Voice characteristics add extralinguistic information on top of the linguistic one and, thus, give information to the listener about the speaker´s individual characteristics, such as age, gender, and physiological and psychological status. This means that when a speaker talks, he/she does not reveal something about self only by what is said but also by how it is said, as speech utterances convey non-lexical interpersonal and discourse related information to a large extent [21–25].

In automatic analysis and speech synthesis, Campbell [22, 23] proposes that speech utterances can be categorized into two main types: 1. the primarily information bearing and 2. the primarily affect-expressing. The first can be characterized by transcriptions alone, but the latter requires knowledge of the speaker´s prosody before an interpretation of the meaning can be made.

## Characteristics of non-verbal communication in speech

Speech prosody has many functions in interpersonal communication, and one of the most important is to indicate changes in speaker´s emotional state. A speaker can express the same

linguistic content with different emotions by changing the prosodic features of speech. These changes in prosody can be intentional or unintentional [24], also, the perceptions of the changes in the prosodic features are subconscious to a large extent [26]. Prosody helps the listener to better understand the discourse [27].

Acoustic parameters that can directly be affected by emotional prosody are mean fundamental frequency (F0) and its variation, mean intensity, and segment and pause duration [e.g., 28, 29]. Changes in arousal level and valence can be reflected in changes in pitch and/or intensity [e.g., 30–32]. Also, epistemic emotions can be categorized by their valence and activation, as enjoyment and curiosity are experienced as pleasant or positive and are therefore considered as positive activating emotions and associated with high arousal and activation while confusion is considered as negative activating emotion entailing negative valence and activating nature [14]. Additionally, voice quality can be affected by speaker´s emotional state, for example, Laver [19] summarizes that breathy voice is associated with intimacy, whispery voice with confidentiality, harsh voice with anger and creaky voice with boredom. According to previous studies [see e.g., 33–35], anger, joy, and fear are manifested in tense voice and sadness, surprise, enthusiasm, intimacy, and content with breathy voice.

When analyzing classroom interaction between teacher and students, Hämäläinen et al. [36] have found that voice research complements to the research. They state that prosodic features, such as intonation, volume, and pace, can add to the research, since it is important to know how things are said in addition to what is said.

This present study aims to investigate whether voice research can be broadened to the study of learning by searching answers to the following research questions:

RQ1: Are the "aha" and "hmm" moments recognizable through certain characteristics in the speaking voice?

RQ2: What can acoustic voice analyses provide to learning research?

## Material and methods

### Ethical statement

The participants were volunteers and gave their written informed consent for the collected material to be used for research. Material was collected and saved, and the results of the analyses were published respecting the Finnish Data Protection Act (Chapter 5, section 31). The Finnish procedure requires that the researchers and the PI are trained in research ethics and have the competence to perform their own ethical evaluation and to assess the risks. The researchers in this study have all completed the University's training course in research ethics and data management.

The Ethical Committee of Research in Humanities in University of Jyväskylä follows the directives of TENK and provides pre-evaluation statements only for studies that pose a substantial risk to participants. The Finnish National board on Research Integrity (TENK) connects such risks to five conditions: deviation from informed consent, intervening in physical integrity of participants, exceptionally strong stimuli, greater than everyday probability to cause mental harm, or threat to safety of participants or close ones (Publications of TENK 3/2019, ISSN 2490-161X; https://tenk.fi/en/advice-and-materials/guidelines-ethical-review-human-sciences). The permission for the study was not applied from the University's Ethical Committee as this study took place within the frame and conduct of a typical study session and applies the procedures for informed consent, triggering none of the aforementioned risk conditions.

## Selection of participants

Twenty-three pre-service physics teachers from the course "Teaching of Physics at school" answered the preliminary questionnaire and completed the task of conceptual understanding instrument DIRECT [37] translated to Finnish, which measured their conceptual knowledge of direct current electric (DC) circuits. The DIRECT instrument is a research-based multiple-choice questionnaire consisting of 29 questions and these multiple-choice questions are a validated tool for measuring conceptual understanding of DC circuits. The questionnaire included questions about personal information, subjective notions of voice, and Voice Handicap Index (VHI) [38]. VHI score can reflect the subjective assessment of a handicap due to voice disorders and can be seen as a screening tool for distinguishing vocal health and vocal dysfunction between individuals [39, 40] and it has been validated for Finnish speakers (Alaluusua & Johansson 2003 [Unpublished]). The participants were third- or fourth-year students (five females and five males) who had already studied the basics of electricity, but their conceptual understanding was somewhat still developing.

A total of five pairs with the lowest VHI scores were selected to the voice related part of the research. The participants were further divided into pairs based on DIRECT instrument results, using varied pairing (pairs with high and low, high and high, and low and low scores). One pair was female-female, one was male-male, and three pairs were female-male. There were altogether three recorded sessions. One participant attended only the first session. Her partner was moved in with another pair, leaving three pairs and one trio to be investigated in the rest of the study.

## Recordings

The recordings took place at the university on three consecutive Fridays from 10 to 12 o´clock. Before each recording the participants answered a short questionnaire about their voice production (easier than usual—as usual—need more effort than usual) and voice quality (better than usual—usual—worse than usual) based on their notions on that day.

The recordings were carried out with AKG C 111 headsets (AKG Harman, Stamford, USA), the microphone placed at 2 cm from the mouth corner. The input frequency was 44.1 kHz, and Revolabs HD Countryman adapters (Yamaha UC, Inc., Sudbury, USA) and Revolabs HD Dual Channel System 2-Ch (Yamaha UC, Inc., Sudbury, USA) combined to Zoom Livetrak L-12 mixer (Zoom Corp., Tokyo, Japan) were used in the recordings. Each voice was calibrated for SPL by using AZ 8922 digital sound level meter (AZ Instrument Corp., Taichung City, Taiwan) (Fig 1). The recordings were approximately 1 hour and 30 minutes in total duration per pair in each session. All participants wore a surgical face mask due to the Covid-19 situation. Although it has been noted that a face mask can have an effect on voice [e.g., 41, 42] it was imperative to use masks during the recordings. The masks were used on every occasion; therefore, the influence of the mask was the same in every situation throughout recordings and thus the mask should not affect the results as comparisons were made individually within sessions.

After the recordings the participants answered a question on a 100-point VAS line about the stressfulness of the situation (0 = not at all stressful—100 = very stressful).

## The tasks

The teaching sessions were structured for problem-based learning in pairs or small groups. All the pairs were given the problem"Rank the light bulbs in DC circuits in the order of their relative brightness" (Fig 2). During the first two recordings, tasks became more complex per each task. In the third recording, the students were given some of the same tasks as in first two

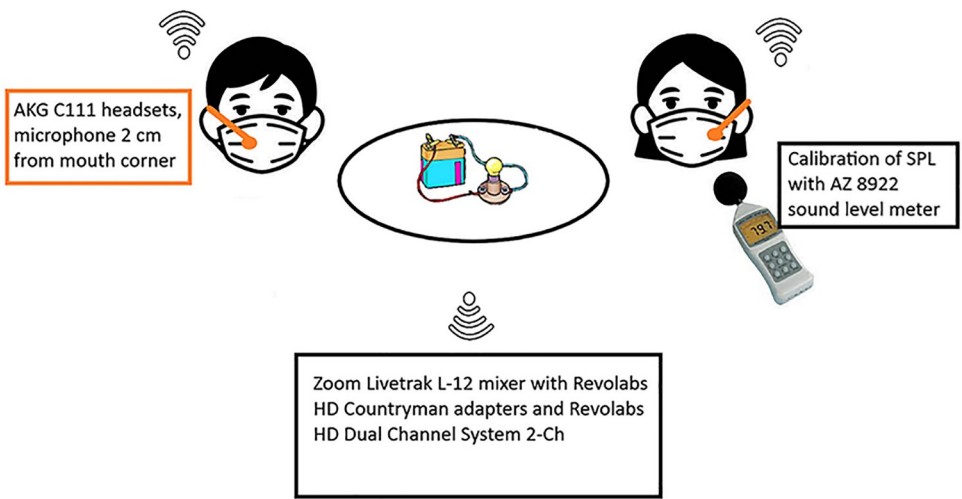

**Fig 1. The recording procedure for one pair.** All pairs were present at the same time in the same location.

recordings—in order to evaluate the progress of learning or understanding that took place. The directions for the problem-based sessions followed the Predict-Observe-Explain structure: First on pen and paper only–analysing the problem first individually and then discussing the prediction as a pair and possibly reaching a consensus–and then constructing the circuits and observing if their prediction was confirmed or if revisions were needed. Based on the observation, the pair formed their final explanation, which was reported to the teacher. The tasks were adapted from McDermott & Shaffer [43]. An example of a task is shown in Fig 2. Even for university students, the tasks are difficult, as they are designed to evoke known misconceptions about electric current, voltage, and resistance. Such misconceptions include assigning properties of energy onto the concept of current [37].

From the recordings, samples indicating either confusion, that is when students expressed uncertainty or impasse in resolving the task(category 1), understanding, i.e. the moments where students expressed an insight of an idea or a conception (2), or explaining, the speech turns where students provided information to peer (3) were extracted by one of the researchers who based the categorization also on context of discussion. Explaining was chosen as one category since it can describe speech and voice with less emotional connotations than understanding and confusion [36]. Also, teachers´ explanative turns can affect students´ learning [44]. Criterium for the samples was that each sample was at least one sentence long and, thus, short interjections or one-word utterances were not included. After categorizing the samples, the ones with overlapping speech, laughter or other disturbances were excluded, and total of 328 analyzable samples were selected for the analysis (94 in category 1, 147 in category 2, and 87 in

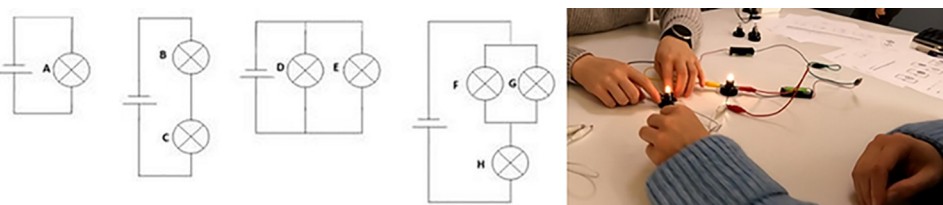

**Fig 2. An example of the task, requesting the student to rank the bulbs A-H per brightness.**

category 3). Additionally, a sample of normal conversational speech was extracted for reference from each session.

For validating the classification of the samples in categories another researcher categorized 55 randomly selected samples, 16.8 percent of the whole, which is an adequate proportion to be multiply coded. Interrater reliability was analyzed by crosstabulation, and the measurement for agreement is Cohen´s Kappa, which estimates the degree of consensus between two judges. Cohen´s Kappa between the two listeners was 0.72 which reads as substantial agreement [45].

## Acoustic analyses

In spoken language, pitch is an important characteristic of voice. It contributes to the perception of intonation in all languages, and to the lexical identity of words in some languages. Pitch contributes with linguistic and paralinguistic functions to the identification of speech acts, the recognition of speaker states, the perception of prosodic structuring, as well as many other characteristics related to discourse and dialogue. A distinction between pitch and fundamental frequency (F0) is that pitch corresponds to the subjective perception of voiced sounds, while F0 corresponds to the physiological parameter of the frequency of vibration of the vocal folds [46]. The most used indicator of the acoustic amplitude of sound wave is sound pressure level (SPL) which correlates with the human perception of loudness [47].

In determining voice quality, two main factors are present: vocal fold vibration and vocal tract resonance, which are controlled by the speaker's phonatory and articulatory behaviors. The voice organ includes the respiratory system, the larynx, and the vocal tract. Movements in the voice organ control the quality of voice sounds [48]. Long-term average spectrum (LTAS) provides information about the sound energy and its distribution to different frequency areas [49] and it is used for voice quality analyses. Alpha ratio is a manifestation of voice quality as it refers to sound level difference between low and high frequency ranges, here the level difference was calculated between 50–1000 and 1000–5000 Hz. Summary of the acoustic parameters and their physiological and perceptual correlates are given in Table 1.

Acoustic analyses of voice provide an objective tool for studying human voice. The acoustic analyses were carried out by Praat 6.0.49 [50] (Fig 3).

Previous studies have shown that the most prominent prosodic features in studying the vocal characteristics during learning are pitch and intensity related characteristics, as well as voice quality and speech tempo [27, 51–54]. The parameters investigated per turn were fundamental frequency (F0), F0 variation and standard deviation (in semitones), sound pressure level (SPL) and its range, Alpha ratio, and pausing (degree of voice breaks in percentages, measured automatically from duration of the breaks in the signal divided by the total duration of the signal). The fundamental frequency was measured in linear Hz scale, but F0 variation and standard deviation of F0 have been expressed in a logarithmic scale (as semitones reference point at 100 Hz), in order to adapt the results to our pitch perception and to make male and female speakers better comparable with each other [46]. An equal (perceived) pitch difference of, say, 2 semitones, comprises different amounts of Hz depending on the actual pitch level. For example, for a male it can be a difference from 98 Hz to 110 Hz (i.e., 12 Hz), while for a

**Table 1. Acoustic parameters and their physiological and perceptual counterparts.**

| Acoustic parameter | Physiologic correlate | Unit | Perceptual correlate |
|---|---|---|---|
| Fundamental frequency (F0) | Frequency of vocal fold vibration | Hertz (Hz) | Pitch |
| Sound pressure level (SPL) | Amplitude of vocal fold vibration | Decibel (dB) | Loudness |
| Long-term average spectrum (LTAS) | Distribution of vocal energy in frequency range | dB/Hz | Voice quality |

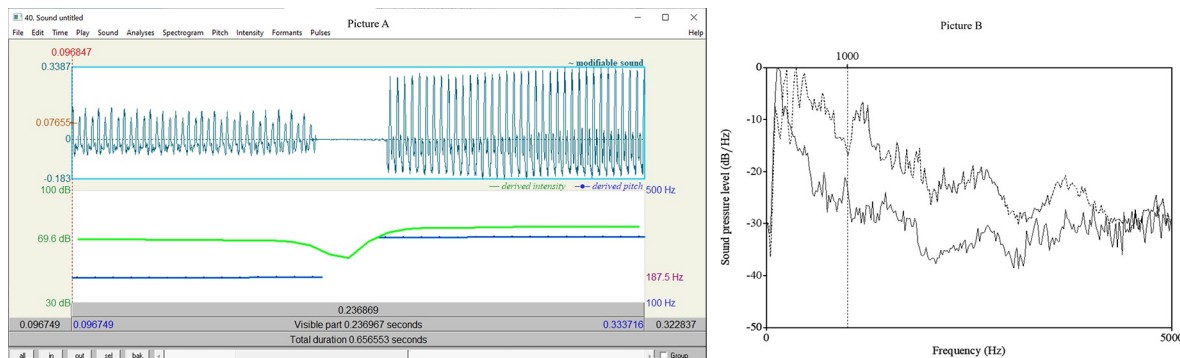

**Fig 3. An example of voice and its analyses in Praat.** Picture A: Female speaker producing vowel /a:/, blue line represents F0, green line SPL. On the left F0 = 188.4 Hz and SPL = 68.7 dB, and on the right F0 = 333.2 Hz and SPL = 76.9 dB. Picture B: LTAS of a female speaker´s speech sample in a 20-second-long text reading, solid line: breathy voice, dotted line: pressed voice.

female it can be from 196 Hz to 220 Hz (i.e., 24 Hz), as female pitch range is on average higher than male. The linear Hertz scale should be transformed to logarithmic scale (e.g., semitones) when differences in frequencies are measured, for example when the span of a speaker´s pitch range or pitch movements are analysed [46]. Changes in vocal loudness can affect the Alpha ratio, since increase in vocal loudness leads to decrease in the overall slope of the long-term average spectrum [55–57], and Alpha ratio is, then, reflecting both voice quality and intensity. In soft and breathy voice, the spectral slope is steep and Alpha ratio low, while in firmer and louder voice the slope is less steep and Alpha ratio bigger (see Fig 3).

The difference between category (confusion, understanding, or explaining) and normal conversational speech was calculated as a subtraction: value in category–value in normal conversational speech. Then, the outcome is a negative value when the value in category is lower than in normal conversational speech. The difference between categories and the normal conversational speech was calculated in percentages, since the amount of the samples varied between the categories and between sexes.

### Statistical analyses

IBM SPSS 26 software was used for the statistical analyses. Kolmogorov-Smirnov test for normality was used to test the distribution of data. Since the distributions were not normally distributed, non-parametric tests were used for comparisons. Friedman test was used for comparisons between categories, Wilcoxon signed ranks test for pairwise comparisons. Significance level was set to .05. Due to the relatively small number of participants, generalized linear mixed model was conducted for finding out possible individual effect in the changes. Significance level was set at .05.

### Results

The participants mean age was 23.8 years (sd 1.93), and the mean VHI score was 5.3 (sd 3.81). Subjective experienced stress was highest in session two. Only one participant evaluated the voice quality to be worse than usual on one occasion (Table 2).

### Recognizing the "aha" and "hmm" moments

Samples for normal speech and explaining had in average longer sample durations than samples in categories of confusion and understanding (Table 3).

**Table 2. Descriptives of participants´ age and VHI scores, experienced stress after each session, and self-estimated voice quality and voice production before each session.**

| | | Age | VHI | Stress Session 1 | Stress Session 2 | Stress Session 3 |
|---|---|---|---|---|---|---|
| Mean | | 23.8 | 5.3 | 29.1 | 38.56 | 35.67 |
| SD | | 1.93 | 3.81 | 18.66 | 16.37 | 18.22 |
| Range | | 21–27 | 0–11 | 0–59 | 10–65 | 5–69 |
| | | | | Session 1 | Session 2 | Session 3 |
| **Voice quality** | better than usual | | | 0 | 0 | 0 |
| | usual | | | 9 | 9 | 8 |
| | worse than usual | | | 0 | 0 | 1 |
| **Voice production** | easier than usual | | | 0 | 0 | 0 |
| | as usual | | | 9 | 9 | 9 |
| | need more effort than usual | | | 0 | 0 | 0 |

N = 9. Experienced stress after each session (0 = not at all stressful—100 = very stressful).

F0 was 5.80 percent higher in understanding and 0.64 percent in confusion compared to normal conversational speech while in explaining it was 5.13 percent lower. F0 standard deviation was 14.41 percent larger in normal conversational speech than in explaining but in confusion and understanding smaller (4.51 and 4.42 percent respectively) (Fig 4).

SPL was 2.65 percent lower in confusion than in normal conversational speech, in understanding it was 3.05 percent and in explaining 0.17 percent higher than in normal conversational speech (Fig 5).

Alpha ratio was 7.33 percent lower in confusion, while in understanding and explaining it was higher (0.54 and 3.79 percent respectively). Pausing was 12.08 percent lower in understanding, in confusion it was 1.45 percent and in explaining 1.59 percent higher than in normal conversational speech (Fig 6).

Means and standard deviations in each parameter in the three categories are given in Table 4.

Between categories, statistically significant differences were found in all parameters except F0 variation and SPL range (Table 5).

Confusion differed significantly from understanding in F0 (Z = -4.35, p < .001, r = -0.45) and F0 SD (Z = -2.36, p = .018, r = -0.24), SPL (-6.50, p < .001, r = -0.67), Alpha ratio (Z = -2.75, p = .006, r = -0.28), and pausing (Z = -3.17, p = .002, r = -0.32) and from explaining in F0 (Z = -4.52, p < .001, r = -0.49), F0 SD (-3.38, p < .001, r = -0.36), SPL (Z = -3,26, p = .001, r = -0,35), and Alpha ratio (Z = -4.50, p < .001, r = -0.49). Understanding differed from explaining in F0 (Z = -7.17, p < .001, r = -0.77), SPL (-3.53, p < .001, r = -0.38), Alpha ratio (Z = -2.00, p = .045, r = -0.22), and pausing (Z = -3.96, p < .001, r = -0.43) (Table 6, Fig 7).

Generalized linear mixed model did not show significant individual effects on the results, Table 7.

**Table 3. Mean and standard deviation of sample duration in each category.**

| Sample duration (sec) | Normal N = 26 | Confusion N = 95 | Understanding N = 147 | Explaining N = 86 |
|---|---|---|---|---|
| Mean | 15.03 | 6.32 | 6.67 | 11.42 |
| SD | 6.81 | 4.03 | 4.78 | 7.08 |

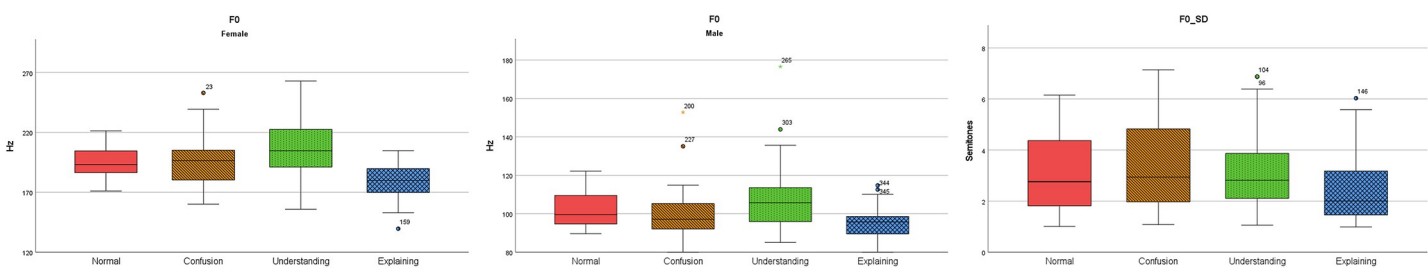

**Fig 4. F0 and F0 SD in normal, confusion, understanding, and explaining.**

## Acoustic voice analyses and learning research

The significant learning moments were manifested differently in the speaking voice and the differences were detectable by acoustic analyses. Acoustic analyses provided significant results between the three categories investigated (Table 5). The results show that acoustic analyses are suitable tool in studying learning moments from learner´s speech and voice.

## Discussion

Speakers modify their prosody for communicative reasons, and for example, by these modifications the same linguistic content can express different emotions [24]. The emotions can be recognized through vocal characteristics [31, 32]. According to Gopnik [7], the "hmm-moments" of learning are closely related to basic emotions of surprise and interest, and the "aha-moments" are accompanied by expression of joy. These moments are, then, often expressed, for example, by facial expression. According to the results in this study, these moments are recognizable also in the speaking voice. In this study, confusion was expressed with softer voice quality than understanding and explaining. Explaining was manifested in lower SPL and alpha ratio (see Figs 5 and 6). This is in line with previous findings that LTAS and Alpha ratio derived from it are affected by voice quality or phonation type (i.e. breathy, modal, or pressed), and SPL [55–57].

Speech samples of this study did not have strong emotional contents, and, thus, it is understandable that differences were found in voice quality related Alpha ratio. In signaling emotions, voice quality and pitch variables may have at least partially different functions: voice quality makes distinctions in general speaker states, moods, and attitudes, whereas pitch variables are more critical in signaling stronger emotions [35]. Understanding can represent stronger emotions, such as excitement [30–32] or joy [7] than confusion or explaining. Therefore, understanding is manifested with higher F0 and louder voice. This is in line with previous studies which suggest that valence and activation are important characteristics of epistemic emotions [14].

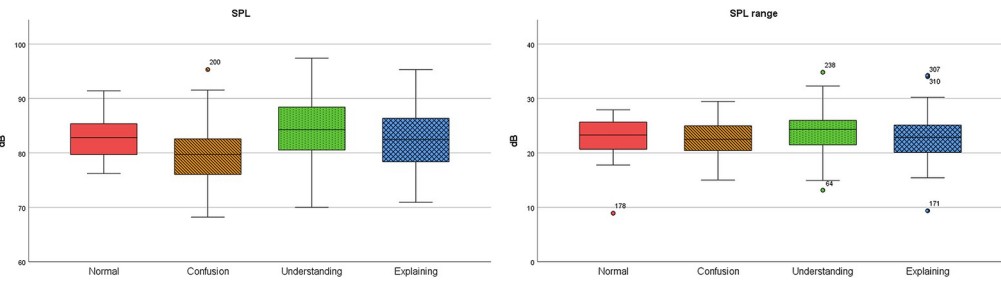

**Fig 5. SPL and SPL range in normal, confusion, understanding, and explaining.**

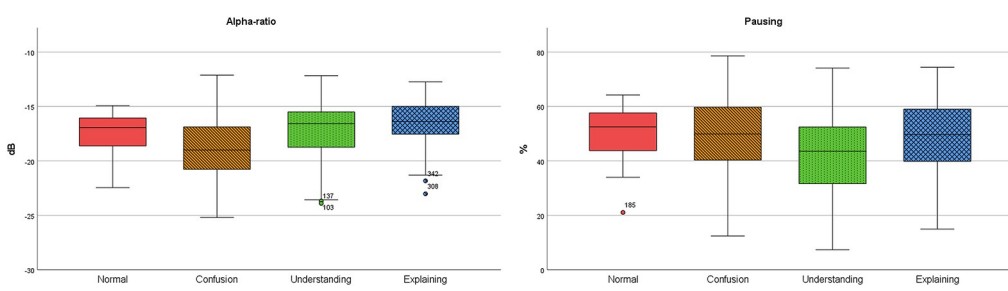

**Fig 6. Alpha ratio and pausing in normal, confusion, understanding, and explaining.**

While explaining had in average longer samples than confusion and understanding, the standard deviation of F0 was smaller. This is in line with previous findings that increase in activity can be manifested with greater pitch variation [58]. Then, both confusion and understanding would have more activity than explaining, as it is possible that when explaining things to another person, the speaker does not express as strong emotional states as in expressing confusion and/or understanding. Hämäläinen et al. [36] found different voice patterns in teacher´s classroom situations. They suggest that pitch variation plays an important role in these patterns, and teacher´s presentative speech has the lowest pitch variation compared to, for example, disputational and promotive speech. Explanatory speech in this study is comparable with presentative speech, and, therefore, the result in our study is in line with their findings. However, further study on acoustic analyses combined with content analysis is required to understand the context and for example, the effects of subject matter understanding on the clear delivery or assertiveness of the speech.

Additionally, understanding had less pausing than confusion or explaining indicating more fluent speech in demonstrating understanding. Speakers can manipulate their use of pausing to structure information, and increased pausing can act as a factor either for cognitive processing for the speaker, or as a communicative-interactive marker giving the interlocutor more time to process the explanatory contents [27, 59, 60]. This was seen in this study in increased degree of pausing in confusion and explaining. In confusion, increased degree of pausing can indicate hesitation, while in explaining it can indicate clarification. For further research, speech rate is worth adding for addressing fluency, hesitation, and clarification markers, since speaking rate tends to increase and pause duration to decrease as uncertainty decreases [27].

Lodge et al. [11] state that, especially when face-to-face interaction is absent, emotions such as confusion can be difficult to detect. As there is a growing use of digital learning environments, a deeper understanding of students´ difficulties should be addressed. This study showed that confusion can be detected from the speaking voice which can provide tools for teachers for recognizing these emotions during, for example, online teaching.

**Table 4. Means and standard deviations (SD) of the difference between normal conversational speech and the three categories, calculated in percentages.**

|  |  | F0 (Hz) | F0 variation (st) | F0 SD (st) | SPL (dB) | SPL range (dB) | Alpha ratio (dB) | Pausing (%) |
|---|---|---|---|---|---|---|---|---|
| **Confusion** | Mean | 0.64 | -5.68 | 4.51 | -2.65 | -0.23 | 7.33 | 1.45 |
| N = 95 | SD | 11.29 | 30.18 | 47.71 | 5.49 | 26.38 | 15.21 | 33.81 |
| **Understanding** | Mean | 5.80 | 3.68 | 4.42 | 3.05 | 6.70 | -0.54 | -12.08 |
| N = 147 | SD | 11.90 | 33.14 | 56.89 | 4.73 | 26.83 | 15.73 | 31.89 |
| **Explaining** | Mean | -5.13 | -5.35 | -14.41 | 0.17 | 8.28 | -3.79 | 1.59 |
| N = 86 | SD | 7.40 | 30.01 | 21.66 | 4.66 | 32.93 | 13.07 | 29.64 |

**Table 5. Comparisons between categories. Friedman test, significance level .05.**

| | F0 (Hz) | F0 variation (st) | F0 SD (st) | SPL (dB) | SPL range (dB) | Alpha ratio (dB) | Pausing (%) |
|---|---|---|---|---|---|---|---|
| **All samples, N = 86, df = 2** | | | | | | | |
| $X^2$ | 62.54 | 1.00 | 10.62 | 44.02 | 2.16 | 14.81 | 15.91 |
| **p** | < .001 | .607 | .005 | < .001 | .34 | < .001 | < .001 |
| **W** | .36 | .006 | 0.06 | 0.26 | 0.01 | 0.09 | 0.09 |

Note: $X^2$ = chi square test statistics for Friedman test, df = degree of freedom, p = statistical significance, W = Kendall´s coefficient for effect size.

Recording speech samples in authentic learning situations set certain challenges to acoustic analyses, since there were several samples that could not be analyzed due to overlapping speech, and other disturbances, such as laughter. However, for gathering authentic samples in the learning dialogues, the procedure was necessary. The inter-rater reliability in categorizing the samples was somewhat moderate which can be explained by the fact that the first categorizing was made from the whole recordings while the other researcher listened to only the selected samples. Therefore, the first categorizing could be influenced by the whole conversation and the contexts where the utterances were spoken.

The participants were compelled to wear surgical face masks during the recordings which can affect especially the spectral characteristics of the voice. According to previous studies [41, 42] wearing a face mask can cause attenuation of the higher frequencies (over 1000 Hz) and thus steeper spectral tilt (which would result in lower alpha ratio), and reduction in intensity. In this study, comparisons were made with individual changes between the normal conversational speech and the categories (i.e., confusion, understanding, and explaining) in each recording session and then, the effect of the mask was the same in each category and also conversational speech. Therefore, the use of a mask should not affect the differences found between the categories.

The number of participants was quite small due to the recording circumstances. However, sample quotient was quite large and, therefore, the statistical analyses can be considered adequate which is confirmed by the effect sizes. Also, the generalized linear mixed model confirms the results as no significant individual effects were found. However, larger number of participants should be included in future research.

**Table 6. Pairwise comparisons between categories.** Wilcoxon signed ranks test, significance level .05.

| | F0 (Hz) | F0 SD (st) | SPL (dB) | Alpha ratio (dB) | Pausing (%) |
|---|---|---|---|---|---|
| **Understanding–Confusion, N = 95** | | | | | |
| **Z** | -4.35 | -2.36 | -6.50 | -2.75 | -3.17 |
| **p** | < .001 | .018 | < .001 | .006 | .002 |
| **r** | -0.45 | -0.24 | -0.67 | -0.28 | -0.32 |
| **Explaining –Confusion, N = 86** | | | | | |
| **Z** | -4.52 | -3.38 | -3.26 | -4.50 | -.56 |
| **p** | < .001 | < .001 | .001 | < .001 | .577 |
| **r** | -0.49 | -0.36 | -0.35 | -0.49 | -0.06 |
| **Explaining –Understanding, N = 86** | | | | | |
| **Z** | -7.17 | -.25 | -3.53 | -2.00 | -3.96 |
| **p** | < .001 | .804 | < .001 | .045 | < .001 |
| **r** | -0.77 | -0.03 | -0.38 | -0.22 | -0.43 |

Note: Z = test statistics for Wilcoxon signed ranks test, p = statistical significance, r = effect size.

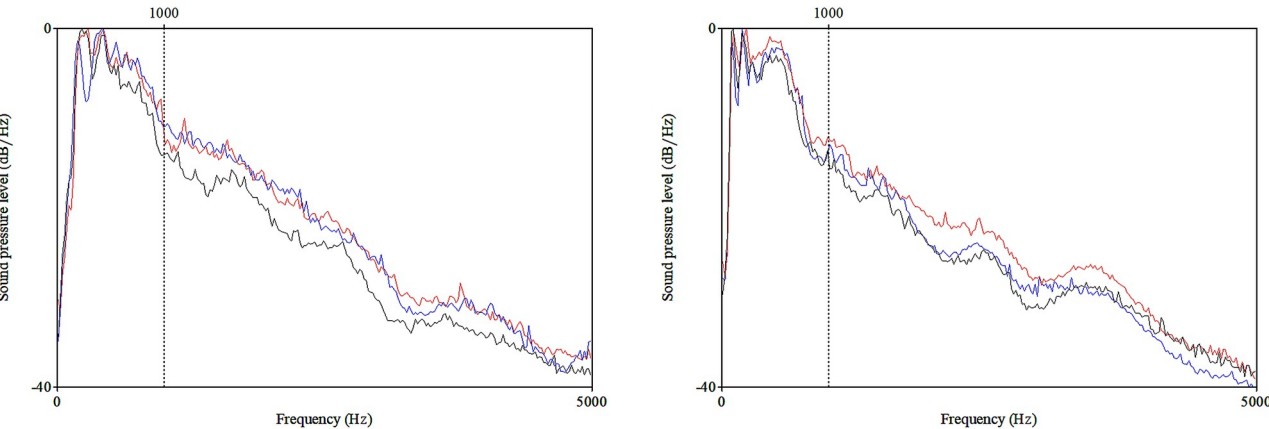

**Fig 7.** LTAS (averaged) for females (left) and males (right). Black line representing confusion, red line understanding, and blue line explaining.

This present study is piloting new research methods for learning research, and for generalizing the results, other situations and environments for the learning discussions should be studied. Additionally, this study was conducted in Finnish cultural and language context. The results may be affected by the cultural use of voice and, thus, the characteristics of voice in other languages and cultural contexts should be investigated. Also, this study concentrated on peer discussions and teacher-student dialogues were not considered. Future research could investigate teacher´s influence on student´ vocal manifestations of the learning moments.

## Recommendations for future work

For finding the learning occasions through speech, further study is required. One possible measure can be the syllabic prosodic index (SPI) introduced by Tavi [24]. This index is a phonetic measure for analyzing prosodic emphasis in syllables, and it combines all main aspects of prosody, that is, pitch, rhythm, and intensity, and it can be used as an addition in analyzing emotional states from voice. SPI could give information about the different ways learning moments are manifested in speakers´ voices. Also, it should be studied whether it is possible to find the crucial moments of learning, or confusion automatically from speech patterns. This would benefit the research in the field of learning since voice parameters can offer objective information regardless of the lexical content of speech and speed up other analysis processes by providing quick access to moments of interest automatically with, for example, data mining and machine learning.

Also, it would be beneficial to study whether machine learning could be prompted to recognize the significant moments of the learning process from students´ speech which could help teachers to detect these moments during online teaching. Next step is to study whether the

**Table 7. Individual speaker effect on parameters.** Generalized linear mixed model, significance level .05.

|  | F0 (Hz) | F0 SD (st) | SPL (dB) | Alpha ratio (dB) | Pausing (%) |
|---|---|---|---|---|---|
| **Fixed effect (Coding)** |  |  |  |  |  |
| **F** | 29.06 | 7.72 | 46.00 | 18.82 | 8.63 |
| **p** | < .001 | < .001 | < .001 | < .001 | < .001 |
| **Random effect (Subject)** |  |  |  |  |  |
| **Z** | 1.40 | 1.82 | 1.79 | 1.76 | 1.55 |
| **p** | >.05 | >.05 | >.05 | >.05 | >.05 |

significant moments can be recognized from audio data by extracting the samples based on such changes in the parameters that were found in this study to be significant in distinguishing different learning moments.

For voice quality analysis, inverse filtering could be added in future research. In inverse filtering [see e.g., 61, 62] glottal flow signal is estimated from corresponding acoustic speech pressure signal. Then, the effect of vocal tract and lip radiation are removed from a microphone signal. Inverse filtering is a method which can reveal more information of voice quality changes between learning instances than spectrum-based Alpha ratio. Also, a larger number of samples and participants are required in future research.

Acoustic analyses provide a possibility to study how voice conveys understanding in the learning process. It should be studied further how speakers´ modify their voice and speech in interaction and how these changes construct understanding.

As significant moments of learning process can be recognized through acoustic analyses of the voice, future research should combine the acoustic analyses with interaction and content analyses. This will give information on emotional reactions in the learning process, for example at different stages of problem-solving or with differences in difficulty of the problems. This novel way of studying the learning process can provide new comprehension of important events of understanding. Also, possible patterns of dissonance, such as frustration, and its manifestation through voice and speech should be studied further.

## Conclusions

Speaker´s voice is affected by the different emotive moments in the learning process which are not necessarily detectable in mere written transcripts. Instances indicating confusion, understanding, and explaining were manifested differently in vocal features. Increase of the activity level may be higher in understanding than in confusion which leads to increase in pitch, loudness, and firmer voice quality in understanding. Explaining was manifested with smaller F0 standard deviation than in confusion. Understanding had less pausing than confusion and explaining. Larger degree of pausing may have reflected hesitation in confusion and served as a tool for clarification in explaining.

We find voice research a promising addition to the traditional research of learning. The successful linking of voice features to moments relevant to the learning process is a promising first start. The work continues as the voice features are studied alongside the conceptual learning progressions of the students. We appreciate voice research for both the ability to triangulate findings from the traditional interaction analyses as well as its standalone value for understanding the way voice is used in the interactive learning situation for multiple purposes.

## Supporting information

**S1 Data.**
(CSV)

## Author Contributions

**Conceptualization:** Kati Järvinen, Anne-Maria Laukkanen, Anna-Leena Kähkönen, Pasi Nieminen, Terhi Mäntylä.

**Formal analysis:** Kati Järvinen.

**Funding acquisition:** Terhi Mäntylä.

**Investigation:** Kati Järvinen, Anne-Maria Laukkanen, Anna-Leena Kähkönen, Pasi Nieminen, Terhi Mäntylä.

**Methodology:** Kati Järvinen, Anne-Maria Laukkanen, Anna-Leena Kähkönen, Pasi Nieminen, Terhi Mäntylä.

**Writing – original draft:** Kati Järvinen, Anne-Maria Laukkanen, Anna-Leena Kähkönen, Pasi Nieminen, Terhi Mäntylä.

**Writing – review & editing:** Kati Järvinen, Anne-Maria Laukkanen, Anna-Leena Kähkönen, Pasi Nieminen, Terhi Mäntylä.

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
