## [Decision Letter · Decision Letter 0]

27 Aug 2024

PONE-D-24-22106Investigating the moments of “aha” and “hmm” through acoustic analysis of voice and speech in pre-service physics teacher education –  A novel method for identifying significant learning momentsPLOS ONE

Dear Dr. Järvinen,

Thank you for submitting your manuscript to PLOS ONE. After careful consideration, we feel that it has merit but does not fully meet PLOS ONE’s publication criteria as it currently stands. Therefore, we invite you to submit a revised version of the manuscript that addresses the points raised during the review process.

We look forward to receiving your revised manuscript.

Kind regards,

Gaoxia Zhu

Academic Editor

PLOS ONE

Journal Requirements:

2. We note that you have referenced (Unpublished master thesis) which has currently not yet been accepted for publication. Please remove this from your References and amend this to state in the body of your manuscript: (ie “Bewick et al. [Unpublished]”) as detailed online in our guide for authors

Additional Editor Comments:

Thank for making the revisions. Please further improve the manuscript by considering the reviewers' comments.

Reviewers' comments:

Reviewer's Responses to Questions

**Comments to the Author**

1. Is the manuscript technically sound, and do the data support the conclusions?

Reviewer #1: Yes

Reviewer #2: Yes

2. Has the statistical analysis been performed appropriately and rigorously? 

Reviewer #1: Yes

Reviewer #2: Yes

3. Have the authors made all data underlying the findings in their manuscript fully available?

Reviewer #1: Yes

Reviewer #2: Yes

4. Is the manuscript presented in an intelligible fashion and written in standard English?

Reviewer #1: Yes

Reviewer #2: Yes

5. Review Comments to the Author

Reviewer #1: The manuscript “Investigating the moments of “aha” and “hmm” through acoustic analysis of voice and speech in pre-service physics teacher education” presents interesting results when analyzing spontaneous speech during interaction of future teachers. Some issues have been cited in previous reviews.

In my opinion, presenting 10 subjects in your analysis is not a demerit. Treating and analyzing spontaneous speech samples is laborious and generates a huge amount of data, presented adequately by the authors, in addition to being a common format in the field of experimental phonetics. However, the data cannot be generalized beyond the study sample and the authors should make this clearer in the discussion, in addition to highlighting the fact that it is valid when checking people wearing masks.

I suggest that in the summary it is clarified that the LTAS measurement is extracted instead of “voice quality through spectrum analysis”, it is clearer for the reader, as with this information alone it is not possible to understand what else is being measured.

It is necessary to make it clear in the method what the authors are referring to when they mention “aha” and when they mention “hmm”. I understand what they are by deduction but in the section “Recognizing the “aha” and “hmm” moments” (page 15) the authors should make this clearer.

I congratulate the authors for the quality of the manuscript and for their work in adapting the revisions requested in previous rounds.

Reviewer #2: This manuscript addresses an interesting aspect related to learning. From what I can understand, the authors addressed many of the concertns raised by the other reviewers.

I suggest to add more detail to the general idea of Problem-based learning, that is cited only at the beginning of tyhje manuscript, with only a single literature reference.

Moreover, the authors should add detail on how they found, in their analysis, the thre categories: confusion, understanding, and explaining. Just saying that they "were extracted by one of the researchers based on context" appears to me too vague. Detail on analysis methods used to extract them is necessary.

6. PLOS authors have the option to publish the peer review history of their article (what does this mean?). If published, this will include your full peer review and any attached files.

Reviewer #1: No

Reviewer #2: No

---

## [Author Response · Author response to Decision Letter 0]

9 Oct 2024

Academic editor´s comments:

Journal Requirements: 

Response: We have ensured this.

2. We note that you have referenced (Unpublished master thesis) which has currently not yet been accepted for publication. Please remove this from your References and amend this to state in the body of your manuscript: (ie “Bewick et al. [Unpublished]”) as detailed online in our guide for authors 

Response: This thesis was necessary to keep, since it validates VHI for Finnish speakers, therefore we moved it to Supporting information (page 32). And amended this to state in the manuscript, line 193.

Response: References have been reviewed and corrected.

Additional Editor Comments: 

Thank for making the revisions. Please further improve the manuscript by considering the reviewers' comments. 

Response: The figures have been uploaded to PACE and ensured they meet the requirements. Figure 7 was remade in color for clearer distinctions.

Reviewers´ comments:

Reviewer's Responses to Questions 

Comments to the Author 

1. Is the manuscript technically sound, and do the data support the conclusions? 

Reviewer #1: Yes 

Reviewer #2: Yes 

2. Has the statistical analysis been performed appropriately and rigorously? 

Reviewer #1: Yes 

Reviewer #2: Yes 

3. Have the authors made all data underlying the findings in their manuscript fully available? 

Reviewer #1: Yes 

Reviewer #2: Yes 

4. Is the manuscript presented in an intelligible fashion and written in standard English? 

Reviewer #1: Yes 

Reviewer #2: Yes 

5. Review Comments to the Author 

Response to reviewers: We thank you for your time and putting your effort into reviewing our manuscript. We find the comments valuable and increasing the merits of our study. Below are our detailed responses to the comments.

Reviewer #1: The manuscript “Investigating the moments of “aha” and “hmm” through acoustic analysis of voice and speech in pre-service physics teacher education” presents interesting results when analyzing spontaneous speech during interaction of future teachers. Some issues have been cited in previous reviews. 

In my opinion, presenting 10 subjects in your analysis is not a demerit. Treating and analyzing spontaneous speech samples is laborious and generates a huge amount of data, presented adequately by the authors, in addition to being a common format in the field of experimental phonetics. However, the data cannot be generalized beyond the study sample and the authors should make this clearer in the discussion, in addition to highlighting the fact that it is valid when checking people wearing masks.

Response: We find this comment very useful. We have now added to the discussion: Lines 455-461: This present study is piloting new research methods for learning research, and for generalizing the results, other situations and environments for the learning discussions should be studied. Additionally, this study was conducted in Finnish cultural and language context. The results may be affected by the cultural use of voice and, thus, the characteristics of voice in other languages and cultural contexts should be investigated. Also, this study concentrated on peer discussions and teacher-student dialogues were not considered. Future research could investigate teacher´s influence on student´ vocal manifestations of the learning moments.

I suggest that in the summary it is clarified that the LTAS measurement is extracted instead of “voice quality through spectrum analysis”, it is clearer for the reader, as with this information alone it is not possible to understand what else is being measured. 

Response: This is a good point. We have now replaced this as suggested, line 25: acoustic voice quality measured by LTAS

It is necessary to make it clear in the method what the authors are referring to when they mention “aha” and when they mention “hmm”. I understand what they are by deduction but in the section “Recognizing the “aha” and “hmm” moments” (page 15) the authors should make this clearer. 

Response: 

Thank you for pointing this out. We have now clarified this. Previous text was: From the recordings, samples indicating either confusion (category 1), understanding (2) or explaining (3) were extracted by one of the researchers based on context. The new and clearer description is now: Lines 242-246: From the recordings, samples indicating either confusion, that is when students expressed uncertainty or an impasse in resolving the task or conception (category 1), understanding, i.e. the moments where students expressed an insight of an idea or a conception (2), or explaining, the speech turns where students provided information to peer (3) were extracted by one of the researchers who based the categorization also on context of discussion. 

Also, we would like to point out that the reliability of the classification was checked through interrater reliability measures, outlined on lines 254-258. 

I congratulate the authors for the quality of the manuscript and for their work in adapting the revisions requested in previous rounds. 

Response: We thank the reviewer for recognizing the merits of the study and appreciating the revisions made for improving the article.

Reviewer #2: This manuscript addresses an interesting aspect related to learning. From what I can understand, the authors addressed many of the concertns raised by the other reviewers. 

I suggest to add more detail to the general idea of Problem-based learning, that is cited only at the beginning of tyhje manuscript, with only a single literature reference. 

Response: We appreciate this comment. We have added more details concerning problem-based learning. 

Lines 42-44: In problem-based learning, the learners orient and analyse the problem, engage in self-directed problem solving, and end with a reporting activity such as explaining their answer to teachers or peers.

Line 48: Added a new reference [2], Yew EH, Goh K. Problem-based learning: An overview of its process and impact on learning. Health professions education. 2016;2(2):75-79. https://doi.org/10.1016/j.hpe.2016.01.004

Lines 226-235: We added problem-based learning when suitable: The teaching sessions were structured for problem-based learning in pairs or small groups. All the pairs were given the problem ”Rank the light bulbs in DC circuits in the order of their relative brightness” (Fig 2). During the first two recordings, tasks became more complex per each task. In the third recording, the students were given some of the same tasks as in first two recordings - in order to evaluate the progress of learning or understanding that took place. The directions for the problem-based sessions followed the Predict-Observe-Explain structure: First on pen and paper only –analysing the problem first individually and then discussing the prediction as a pair and possibly reaching a consensus – and then constructing the circuits and observing if their prediction was confirmed or if revisions were needed. Based on the observation, the pair formed their final explanation, which was reported to the teacher.

Lines 491-492: This will give information on emotional reactions in the learning process, for example at different stages of problem-solving or with differences in difficulty of the problems.

Moreover, the authors should add detail on how they found, in their analysis, the thre categories: confusion, understanding, and explaining. Just saying that they "were extracted by one of the researchers based on context" appears to me too vague. Detail on analysis methods used to extract them is necessary. 

Response: We appreciate this comment. We have now strengthened the categorization of the samples. Previous text was: From the recordings, samples indicating either confusion (category 1), understanding (2) or explaining (3) were extracted by one of the researchers based on context. The new and clearer description is now, Lines 242-246: From the recordings, samples indicating either confusion, that is when students expressed uncertainty or an impasse in resolving the task or conception (category 1), understanding, i.e. the moments where students expressed an insight of an idea or a conception (2), or explaining, the speech turns where students provided information to peer (3) were extracted by one of the researchers who based the categorization also on context of discussion. 

Also, we would like to point out that the reliability of the classification was checked through interrater reliability measures, outlined on lines 254-258. 

6. PLOS authors have the option to publish the peer review history of their article (what does this mean?). If published, this will include your full peer review and any attached files. 

Do you want your identity to be public for this peer review? For information about this choice, including consent withdrawal, please see our Privacy Policy. 

Reviewer #1: No 

Reviewer #2: No

---

## [Decision Letter · Decision Letter 1]

11 Nov 2024

Investigating the moments of “aha” and “hmm” through acoustic analysis of voice and speech in pre-service physics teacher education –  A novel method for identifying significant learning moments

PONE-D-24-22106R1

Dear Dr. Järvinen,

We’re pleased to inform you that your manuscript has been judged scientifically suitable for publication and will be formally accepted for publication once it meets all outstanding technical requirements.

Kind regards,

Gaoxia Zhu

Academic Editor

PLOS ONE

Additional Editor Comments (optional):

Thanks for your efforts in addressing the reviewers' comments and improving the manuscript. Both reviewers indicated that their concerns had been addressed and recommended acceptance of the submission. Congratulations!

Reviewers' comments:

Reviewer's Responses to Questions

**Comments to the Author**

1. If the authors have adequately addressed your comments raised in a previous round of review and you feel that this manuscript is now acceptable for publication, you may indicate that here to bypass the “Comments to the Author” section, enter your conflict of interest statement in the “Confidential to Editor” section, and submit your "Accept" recommendation.

Reviewer #1: All comments have been addressed

Reviewer #2: All comments have been addressed

2. Is the manuscript technically sound, and do the data support the conclusions?

Reviewer #1: Yes

Reviewer #2: Yes

3. Has the statistical analysis been performed appropriately and rigorously? 

Reviewer #1: Yes

Reviewer #2: Yes

4. Have the authors made all data underlying the findings in their manuscript fully available?

Reviewer #1: Yes

Reviewer #2: Yes

5. Is the manuscript presented in an intelligible fashion and written in standard English?

Reviewer #1: Yes

Reviewer #2: Yes

6. Review Comments to the Author

Reviewer #1: The authors answered my questions satisfactorily and made some inclusions that clarified the points of concern. The points were addressed during the discussion and in some points of methods.

Reviewer #2: (No Response)

7. PLOS authors have the option to publish the peer review history of their article (what does this mean?). If published, this will include your full peer review and any attached files.

Reviewer #1: No

Reviewer #2: No

---

## [Editor Report · Acceptance letter]

14 Nov 2024

PONE-D-24-22106R1 

PLOS ONE

Dear Dr. Järvinen, 

I'm pleased to inform you that your manuscript has been deemed suitable for publication in PLOS ONE. Congratulations! Your manuscript is now being handed over to our production team.

Kind regards, 

on behalf of

Dr. Gaoxia Zhu 

Academic Editor

PLOS ONE